# Impact of adverse events during community-wide mass drug administration for soil-transmitted helminths on subsequent participation–a Theory of Planned Behaviour analysis

**Kumudha Aruldas[1‡], Gideon John Israel[1‡], Jabaselvi Johnson[1], Angelin Titus[1], Malvika Saxena[1], Saravanakumar Puthupalayam Kaliappan[1], Rohan Michael Ramesh[1], Judd L. Walson[2,3], Arianna Rubin Means[3,4], Sitara S. R. Ajjampur[1]***

**1** The Wellcome Trust Research Laboratory, Division of Gastrointestinal Sciences, Christian Medical College, Vellore, India, **2** Departments of Global Health, Medicine, Pediatrics and Epidemiology, University of Washington, Seattle, Washington, United States of America, **3** The DeWorm3 Project, University of Washington, Seattle, Washington, United States of America, **4** Department of Global Health, University of Washington, Seattle, Washington, United States of America

‡ Co-lead authors
* sitararao@cmcvellore.ac.in

**Data Availability Statement:** Some restrictions will apply because there may be some sensitivity and

## Abstract

### Background

Experiencing adverse events (AEs) during mass drug administration (MDA) could affect participation in future MDAs. This study aims to understand the potential influence of AEs during a community-wide MDA (cMDA) trial for soil-transmitted helminths (STH) in India on intention to participate in future cMDAs.

### Methods

This study was conducted using a multi-method quantitative and qualitative approach among 74 participants who experienced an AE during STH cMDA and the 12 participants who subsequently refused cMDA treatment of the ongoing DeWorm3 trial. Path analysis and thematic analysis guided by the Theory of Planned Behaviour, was used.

### Principal findings

Among 74 individuals who reported an AE, 12% refused treatment in the cMDA immediately subsequent to their AE and 4% refused in all subsequent cMDAs. Of these 74 individuals, 59 (80%) completed a survey and eight participated in in-depth interviews. A positive attitude towards deworming and perceived ability to participate in cMDA (perceived behavioural control) were significant predictors of intention to participate in cMDA ($p<0.05$). A positive attitude towards cMDA was associated with caste ($\chi^2 = 3.83$, $P = 0.05$), particularly among the scheduled caste/scheduled tribe (SC/ST) (62%). Perceived behavioural control in cMDA participation was associated with occupation ($\chi^2 = 5.02$, $P<0.05$), with higher

potentially identifiable information where participants describe individual-level experiences. All relevant data are within the manuscript and its supporting information files.

**Funding:** The DeWorm3 Project is funded by a grant from the Bill & Melinda Gates Foundation (OPP1129535, PI JLW). The funders had no role in study design, data collection, analysis, decision to publish, or manuscript preparation.

**Competing interests:** The authors have declared that no competing interests exist.

perceived control among those engaged in skilled occupations (78%). Intention to participate in subsequent cMDAs was associated with caste and family type ($\chi^2 = 3.83$, $P = 0.05$ and $\chi^2 = 7.50$, $P<0.05$ respectively) and was higher among SC/ST (62%) and those with extended families (67%). In-depth interviews demonstrated that perceived severe AEs may lead to treatment refusal in future, particularly if children were affected.

## Conclusions

Intention to participate in future STH cMDAs was associated with caste (SC/ST) and family type (extended families). Therefore, community mobilization messages about potential AEs and their management may need to intentionally target non-SC/ST households, nuclear families, and those engaged in unskilled occupations to increase cMDA participation given the possibility of AEs occurring.

## Trial registration

NCT03014167, ClinicalTrials.gov.

## Author summary

This paper sampled individuals who experienced an AE during community-wide MDA (cMDA) with albendazole for soil-transmitted helminths in the DeWorm3 trial in India to understand their intention to participate in future cMDAs. This study utilized the theory of planned behaviour, which postulates that an individual's intention to change a behaviour depends on their attitude, perceived expectations of people around them (social norms), and their perceived behavioural control to adopt the behaviour (self-efficacy). This study indicated that positive attitude towards cMDA and their perceived behavioural control in cMDA participation positively influences their intention to participate in future cMDAs. Other factors associated with intention to participate in cMDA were caste, family structure, occupation, and participation in community sensitisation activities. In-depth interviews revealed that while many individuals participate in subsequent cMDAs after an AE because of the presumed health benefits of deworming, some are likely to refuse treatment due to fear of AEs, particularly fears of AEs among children. To increase participation rates in cMDA programs where AEs undoubtably do occur, targeted counselling of groups at risk of non-participation and assuring care and support during AEs may be important strategies to improve coverage.

## Introduction

India is estimated to have about 258 million people infected with soil-transmitted helminths (STH) including *Ascaris lumbricoides*, *Trichuris trichiura*, and *Ancylostoma duodenale* or *Necator americanus* (hookworms) [1]. The World Health Organization (WHO) Roadmap 2030 for Neglected Tropical Diseases (NTD) includes the goal to eliminate STH-related morbidity (usually caused by moderate to heavy intensity infections, MHI) by ensuring 75% coverage in targeted deworming of vulnerable populations including pre-school aged children (PSAC) 1–4 years), school-age children (SAC) (5–19 years) and women of reproductive age (15–49 years) [2]. Mathematical modelling and the 'TUMIKIA' field trials indicate that

expanded, community-wide mass drug administration (cMDA) for deworming could potentially interrupt STH transmission [3–6]. In this context, a multi-country community-based cluster-randomized trial, DeWorm3, is underway in Benin, India, and Malawi to assess the feasibility of interrupting STH transmission with six rounds of biannual cMDA of albendazole [7–9].

Adverse events (AEs) experienced during treatment in MDAs and the fear of side-effects to the treatment can often lead to refusals to participate in MDA programs [10–14]. Further, not receiving care for suspected AEs experienced during previous MDAs can also deter community members from participating in future MDAs [15]. In India, albendazole has been co-administered with diethylcarbamazine in Lymphatic Filariasis (LF) MDA programs since 2007. Albendazole is also administered biannually in the country's school-based deworming program, one of the largest in the world, targeting ~240 million children [16,17]. Several LF-MDA studies in India have reported AEs in the range of 3–9% such as dizziness, drowsiness, nausea, vomiting, and fever, however, they are not specific to albendazole intake alone [10,11,13,18,19]. There are limited data concerning participant behaviours after an AE, and how experiencing an AE influences willingness to participate in future rounds of cMDA.

The Theory of Planned Behaviour (TPB) predicts an individual's intention to modify or adopt a behaviour. The TPB describes three predictors of human intention to adopt a behaviour or action, such as participating in cMDA after experiencing an AE [20]. One is participant's attitude, the individual's beliefs about the expected positive or negative outcomes of the behaviour or action; second is subjective norms, an individual's perceived social or peer expectations of their behaviour or action; and third is perceived behavioural control, the individual's beliefs about their ability or self-efficacy to adopt the behaviour or perform the action. Change in any of these three predictors account for considerable variance in individual's intention to adopt a behaviour. TPB specifically includes perceived behavioural control as factors like money, time or other resources, and control by family member are barriers to individual's ability to access a service or adopt a behaviour. This theory provides a conceptual framework that can be applied to any behaviour of interest.

Very few studies have used the TPB to assess participation or uptake of interventions for infectious diseases. These include assessment of interventions such as sleeping under an insecticide-treated mosquito net for malaria prevention in Iran and adherence to tuberculosis treatment in Indonesia [21,22]. TPB has been used to evaluate intentions to adopt COVID-19 related behaviour changes as preventive measures during the pandemic, such as maintaining physical distancing, wearing a mask, and vaccination uptake in a range of countries including the USA, China, Pakistan, and Israel [23–28]. This multi-method study in the state of Tamil Nadu, India, aimed to use the TPB framework to understand perceptions about participating in cMDA for STH and drivers of participation in subsequent STH cMDA programs after experiencing an AE.

## Methods

### Ethics statement

The Institutional Review Board (IRB) of Christian Medical College (CMC), Vellore (10392 [INTERVEN]), and the Human Subjects Division at the University of Washington (STUDY00000180) approved the DeWorm3 trial in India. The trial is registered at Clinical-Trials.gov (NCT03014167). The CMC IRB approved this study on AE among Deworm3 trial participants as an amendment (IRB–A12, October 28, 2020). Written informed consent was obtained from all who participated in the survey and in-depth interviews. In case of children,

the adult primary caregivers of the children (parent/guardian) who were present when the child experienced the AE were interviewed with written informed consent.

Reporting of this study has been verified in accordance with the Strengthening the Reporting of Observational Studies in Epidemiology (STROBE) checklist (S1 File).

## Study location and setting

The DeWorm3 trial site in southern India is located in two blocks of Tamil Nadu—the Timiri block in Ranipet district (formerly Vellore district) and the Jawadhu Hills block in Tiruvannamalai district. In 2017, at the start of the trial, 140,932 individuals were enumerated in the study area residing in 6,536 households across 373 villages. The socio-demographic details of the study population and the baseline STH prevalence have been previously published [7,29]. The population was demarcated into 40 clusters and randomized to 20 intervention and 20 control clusters [8]. During the cMDA in the intervention clusters, the community drug distributors (CDDs) distributed albendazole door-to-door to all community members in the ages of 1–99 years excluding women in their first trimester of pregnancy, those with history of adverse reaction to benzimidazoles, and those acutely ill at the time of cMDAs [8]. In the control clusters, the Ministry of Health and Family Welfare (MoHFW) implemented routine biannual school-based treatment of PSAC and SAC on National Deworming Days [16]. During cMDA, field workers and supervisors engaged in the trial, accompanied the CDDs to capture real-time treatment consumption data using a mobile phone application. The trial has completed six rounds of biannual cMDA during 2018–2020 and is currently in the post-intervention surveillance phase.

During and for seven days following the cMDAs, passive surveillance of AEs was conducted where community members reported their AEs to the CDDs or the DeWorm3 field workers. The CDDs informed the field workers who in turn reported the AEs immediately to the respective medical teams set up for the trial in the two study blocks. The medical teams visited households where an AE was reported, documented the details of the AE, and provided treatment, follow-up care, and support.

## Study design and data collection

A multi-method approach was used, including quantitative and qualitative data collection. The medical teams recorded 75 AEs among 74 participants (one participant reported AEs in two rounds of cMDA) during the six rounds of cMDA. All 74 individuals who reported an AE were invited six months after completing the last cMDA to complete a survey during March-April 2021. For AEs reported among the 14 children under 15 years of age, the primary caretakers (parent or guardian) present at the time of their AE participated in the survey on behalf of their children. In parallel, eight in-depth interviews were conducted with individuals who reported an AE and also refused treatment in one or more of the subsequent cMDA rounds. Primary caregivers of children (two out of eight) were interviewed on behalf of the children.

The participation in STH cMDA is a voluntary preventive health behaviour, therefore, the TPB constructs were used to guide the development of the questionnaire, qualitative guidelines, and data analysis [20]. Quantitative data were collected via a 37-item questionnaire consisting of 15 questions about background variables, 19 questions with 5-point Likert scale questions ('strongly agree' to 'strongly disagree') and three questions with 3-category nominal scale questions ('yes', 'not sure', and 'no') to measure the three independent variables (attitude, subjective norms, perceived behavioural control) and outcome variable (intention to participate) (S2 File). The questionnaire was translated into Tamil (local language), piloted, and administered by four trained research assistants.

The in-depth interviews were conducted using a semi-structured guideline (S2 File) to understand participant's experiences of an AE, why they declined subsequent treatment and their perspective on participating in future STH cMDA programs. The in-depth interviews were conducted in the local language (Tamil) by two trained social science research assistants using an interview guide and were audio-recorded.

## Data analysis

We hypothesized that individuals with favourable beliefs towards participating in cMDA would have a stronger intention to participate in future cMDA rounds, even after reporting an AE. The responses to five-point Likert scale questions of 'strongly agree' to 'strongly disagree' were scored as 5 to 1, with 5 being the most positive response and 1 being the most negative response. The responses to the three category nominal scale questions of 'yes', 'not sure', and 'no', were scored as 5, 3, and 1, respectively. All questions were given equal weightage. The individual scores of the questions in each TPB construct were added to arrive at the composite score. The scores equal to and above the median of the composite scores were categorised as a 'high' or 'positive' construct and below-median as a 'low' or 'negative' construct [30].

The internal consistency of the questionnaire was tested using Cronbach's alpha for reliability [31]. The questions for measurement of attitude (eight questions), perceived behavioural control (six questions), subjective norm (four questions), and intention to participate in future cMDAs (four questions) had an alpha value of 0.65, 0.64, 0.44 and 0.74, respectively. The 15 independent variables were respondent age, gender, marital status, caste, religion, education, occupation, family structure, highest level of household education and occupation, household toilet, membership in any community-based organizations, exposure to community sensitization activities, familiarity with the CDD carrying out cMDA in their neighbourhood, and perceived severity of AEs. Descriptive statistics were used to report the total number and proportion of individuals in each demographic group. Chi-Square and Fisher's exact test, significant at $P<0.05$, was used to determine if there are associations between the background variables of interest and the three independent variables and outcome variable of TPB. We used path analysis to examine the relationship between a dependent variable and two or more independent variables to evaluate causal models [32]. Path analysis determines the degree and significance of the causal relationship between variables. The path analysis calculates all the paths simultaneously and yields overall goodness of fit measure for the model. In this analysis, the structural path model, a type of directed graph, was formed by linking the independent variables (attitude, subjective norm, and perceived behavioural control) to the outcome variable (intention to participate), where each independent variable is hypothesized to influence the outcome variable. The data were analysed with STATA 16.0 software (StataCorp, TX, USA) and Gpower 3.1 was used to calculate the power of a statistical test for the path analysis [33]. The statistical power (0.97) was computed using an effect size ($f^2$) of 0.35 (calculated to be large using Cohen's $f^2$), at $\alpha = 0.05$, with n = 59, and three predictors. The audio-recorded in-depth interviews were transcribed verbatim and translated into English by the two interview facilitators. The transcripts were coded by two primary coders independently using a deductive approach using an *a priori* TPB-based codebook, the discrepancies in coding between the two primary coders were reviewed and revised with consensus and analysed in ATLAS.Ti 8.0.

## Results

### Experience of AEs and participation in subsequent cMDAs

A total of 74 participants reported related or unrelated AEs during the six rounds of STH cMDA. Of these, three (one child and two adult women) were determined as serious AEs by

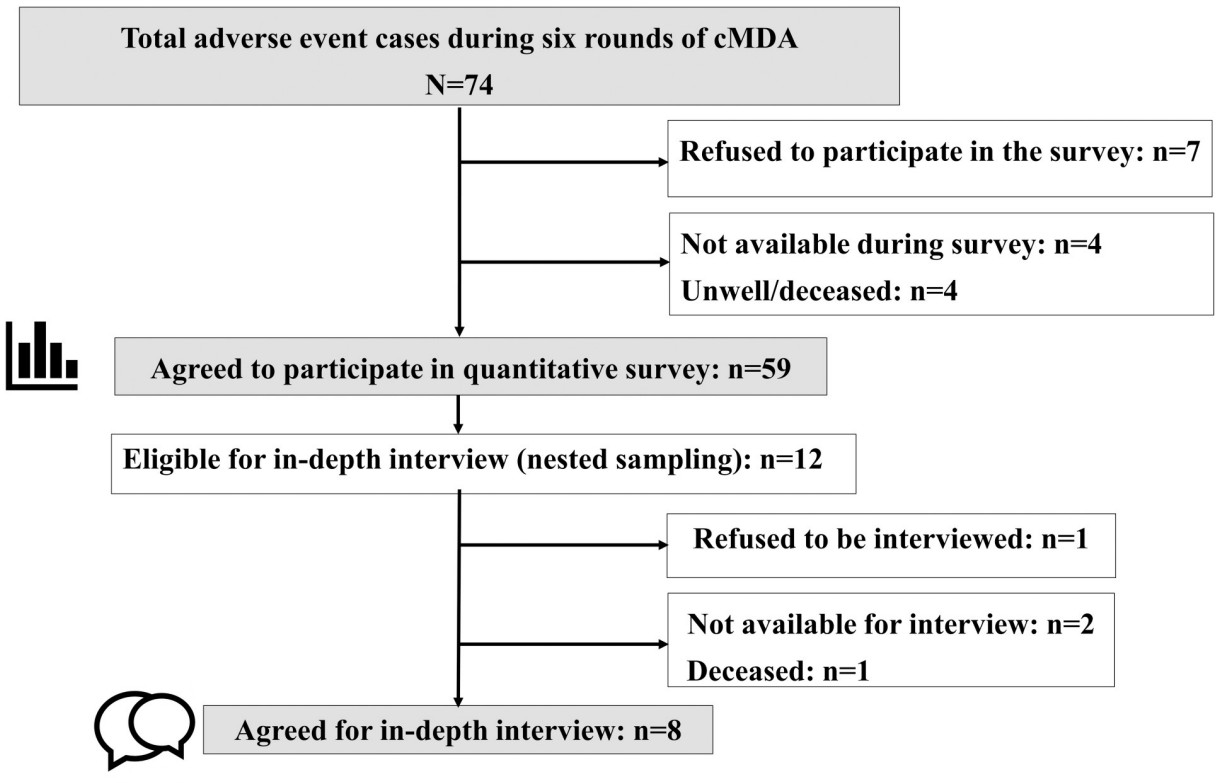

**Fig 1. AE study participation chart.**

the medical team but unrelated to albendazole. Overall, 8% of the AEs recorded were in PSAC, 14% were in SAC, 22% were in men, and 57% were in women above the age of 15 years. The AEs recorded by the medical teams based on the complaints reported by the participants were diarrhoea (49, 66.2%), gastritis (8, 10.8%), skin allergy (6, 8.1%), fever (5, 6.8%), stomach pain (3, 4.1%), cough (1, 1.4%), dizziness (1, 1.4%), and urinary tract infection (1, 1.4%).

Fifty-nine of the 74 eligible participants consented to participate in the survey on AE (80%) (Fig 1). Of the 59 survey participants, 43 were women (73%), and 16 were men (27%). They were between the ages of 19–72 years (median, interquartile range (IQR): 45.0, 29.0–56.0). A majority were Hindu (93%), many were married (75%), and a little over half belonged to a disadvantaged socio-economic group listed in a schedule of the Indian constitution as scheduled caste or scheduled tribe (58%) (SC/ST) and lived in an extended family (includes siblings and their families or has at least three generations) (56%). Less than half were educated to the level of secondary school or more (34%), however, 80% had someone in their family who was educated to the level of secondary school or more. Many of the participants had someone in the family who was a member of either a women's self-help group, a farmers' club, or a cooperative society (44%). About half of them had attended DeWorm3 community sensitization activities conducted prior to a cMDA (49%), and many were familiar with the CDDs delivering cMDA in their village (88%). Of the 15 individuals who did not participate in the survey, 2 (13%) were children aged 1–15 years, 8 (54%) were women, and 5 (33%) were men. They did not feel the need to participate because their AE had resolved and were feeling well.

Of the 74 who experienced an AE, 84% consumed the tablet in all subsequent rounds of cMDA, 12% refused treatment in the immediate subsequent round, and 4% refused treatment in all subsequent rounds (Fig 2). Among the 12 who refused to treatment in one or more

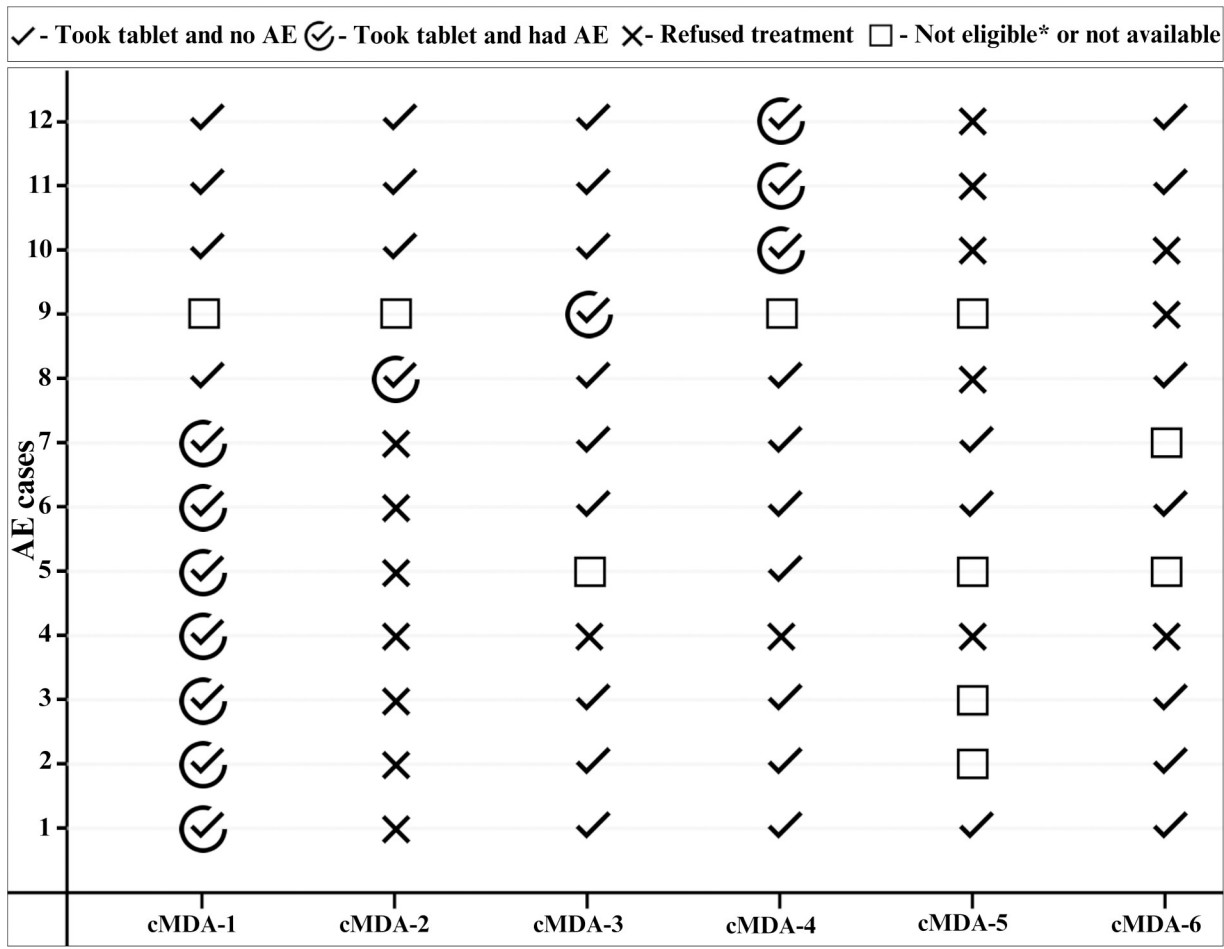

**Fig 2. Treatment history of 12 participants who refused treatment in subsequent rounds of cMDA after experiencing an AE.** * Not eligible–seriously ill or received deworm treatment within two weeks prior to cMDA.

subsequent rounds of cMDA, eight out of the nine participants consented to an in-depth interview. One elderly man did not participate because his family members did not want him to be disturbed by an interview.

### CDDs were the first person most participants contacted when they experienced an AE, and were satisfied with their response

Most of the participants were familiar with the CDDs who distributed the albendazole to them during the cMDAs (88%). CDDs were the first person the participants contacted when they experienced an AE (81%) and were 'very satisfied or satisfied' with the advice given and actions taken by the CDD (94%). As a next step, the CDDs informed the DeWorm3 field workers about the AEs and coordinated the house visits of medical teams. The in-depth interviews also revealed that most respondents were familiar with the CDDs and some even knew them by their names because the CDDs were from the same village. The participants contacted the CDDs immediately if the symptoms occurred on the same day after treatment and when they perceived that the symptom was due to the tablet. Those who did not attribute the cause of the symptom to the treatment waited for two days before informing the CDDs. The participants were satisfied with CDDs because they had visited them immediately after the AE was reported and subsequently initiated house visits and follow-ups of medical teams. They said,

"*She (CDD) came immediately from next street, within 5 minutes. . . she called the doctor on phone. . .. Doctor also came. . .. Kept coming for three days. . . and spoke well. Doctor asked me to call immediately if something happens, and they will come immediately. . . Yes, I was satisfied.*" (Woman #1- perceived moderately severe AE, Cluster 23)

"*I went to see her (CDD) and asked her, "What tablet did you give me?. . . Immediately, she called those who came along with her to distribute the tablet. They came immediately. . . doctor was one among them, they gave me a tablet, after eating that tablet I got well. . . To a great extent I am satisfied. . . Is it not good when things happen in a good manner*?" (Man #1- perceived severe AE, Cluster 13)

## Despite experiencing an AE, most believed that the tablet was safe and effective, and deworming was necessary

Among the 59 participants surveyed, 58% perceived their AE as 'severe or very severe', 24% reported 'moderate' severity and 18% reported 'mild or very mild' severity. Most of the participants attributed their AE to multiple causes, including albendazole distributed during cMDA (85%), worms they had in their stomach (67%), food they had consumed that day (22%), other associated illnesses (5%), or to consuming the tablet on an empty stomach (4%). The in-depth interviews also showed that some of them strongly believed that the AE occurred directly and solely due to the tablet, as they said,

"*Yes, only due to that tablet. . .I had suffered that sort of diarrhoea,. . . it happened because of eating that tablet. . .. I did not have any other problem, it happened by eating the tablet.*" (Woman #1- perceived moderately severe AE, Cluster 24)

"*I ate only dhal and rice on that day, which is usually what I eat. For indigestion to happen, did I eat meat and rice, to cause it? Only on that day. . . This happened only after eating this tablet.*" (Man #1- perceived severe AE, Cluster 13)

Despite attributing the AE to the tablet distributed during cMDAs, a majority indicated that deworming was good for their health (93%), and the drug distributed in cMDAs was safe and effective (85%). Most of them believed that those who do not have symptoms of STH should also be dewormed (88%) but about half of them thought that deworming was not necessary for those who use toilets (53%). Many of them believed that STH is only a minor infection (49%) but believed that the adults were also affected (80%). During the in-depth interviews, they explained this as,

"*Everyone has intestinal worms. It is not that intestinal worms affect only adults or children, but intestinal worms do exist in excess for children. . . Intestinal worms come out automatically when adults eat tablet for purging and cleansing. . . We do not know whether in our stomach intestinal worms exist or not.*" (Man #1- perceived severe AE, Cluster 13)

"*It (STH) is not a minor one. . . It will become a lump. . . cause trouble to the body, every month menstruation will not be regular,. . . if we eat the tablet, menstruation happens correctly. . . Infection can start with adults later transmit to children as well.*" (Woman #1- perceived moderately severe AE, Cluster 23)

In-depth interviews among those who had refused to participate in at least one cMDA, showed that they perceived their AE to be mild to moderately severe. Most of them reported

that their symptoms appeared within a few hours of ingesting the tablet and lasted for about 1–3 days. Fear of an AE was also observed irrespective of their perceived severity of an AE. Fear of severe AEs among children can prevent them from participating in future cMDAs. They said,

> "*I am eating the tablet which drug distributors give. . .. I strongly believe it is important for me. I tell my family members to eat. After my grandson's sickness which happened after he ate the tablet, I cannot force my family to eat the tablet. I think eating deworm tablet is important.*" (Grandfather- perceived very severe AE, Cluster 15)

> "*Six months ago, they gave the tablet and I had diarrhoea. When they came again to give the tablet, I said no to tablet, rest of my family members also did not eat the tablet. No one ate the tablet. Same fear.*" (Man #1- perceived severe AE, Cluster 13)

Overall, 51% of individuals had a positive attitude (≥median score of 34; maximum score–40) towards cMDA, despite having reported a previous AE. Positive attitude towards STH cMDA was significantly associated with caste ($\chi^2$ = 3.83, *P* = 0.05) (Table 1). A higher proportion of those who belonged to SC/ST caste (62%) had a positive attitude towards cMDA for STH than those who belonged to other castes (36%).

## Individuals with strong ties to family and community are more likely to believe that those around them are participating in cMDA

Most of the participants believed that their neighbours and friends participated in cMDAs (85%), and that people in their community considered it important to control or stop the spread of STH (73%). Many participants thought that their neighbours and friends would expect them to participate in cMDAs (63%), and few of them felt that their participation will be influenced by their neighbours' and friends' participation (10%). These findings corroborated with results of the in-depth interviews, where most participants reported that everyone in their families, as well as people in their village, were taking the tablet and continued to take it because of its health benefits. They said,

> "*People do think of completely eradicating (STH). Everyone in the village thinks that these intestinal worms would not be generated again by eating this tablet. Those who eat this tablet will benefit from it.*" (Woman #1- perceived moderately severe AE, Cluster 38)

> "*I eat the tablet when it is given to me, similarly others also say. . .. Yes, neighbours ask. . . They ask, "Did you get the tablet and eat?" I say that I did take the tablet and eat.*" (Woman #3- perceived moderately severe AE, Cluster 38)

A high score for subjective norm indicates individuals' perception that the people around them participate in cMDAs and would like to control or stop STH transmission. Overall, 51% of the participants showed a high score for subjective norm (≥median score of 18; maximum score–20) for STH cMDA. A high subjective norm score was significantly associated with family type and prior participation in DeWorm3 community sensitisation activities ($\chi^2$ = 7.50, *P<0*.05; and $\chi^2$ = 4.91, *P<0*.05, respectively) (Table 1). Among those who belonged to extended families, 67% had high subjective norm scores compared to those who belonged to nuclear families (31%). Among those who participated in DeWorm3 community sensitisation activities, 66% had high subjective norm scores compared to those who did not (37%).

**Table 1. Factors associated with attitude, subjective norm, perceived behavioural control, and intention to participate in future STH cMDAs.**

| Factors | Total | Attitude | | | Subjective norm | | | Perceived behavioural control | | | Intention to participate | | |
|---|---|---|---|---|---|---|---|---|---|---|---|---|---|
| | n = 59 | Low | High | Chi-Square test | Low | High | Chi-Square test | Low | High | Chi-Square test | Low | High | Chi-Square test |
| | | n = 29 | n = 30 | | n = 29 | n = 30 | | n = 26 | n = 33 | | n = 29 | n = 30 | |
| | n (%) | n (%) | n (%) | p-value | n (%) | n (%) | p-value | n (%) | n (%) | p-value | n (%) | n (%) | p-value |
| **Gender** | | | | | | | | | | | | | |
| Male | 16 (27.12) | 11 (68.75) | 5 (31.25) | 0.066 | 8 (50.00) | 8 (50.00) | 0.937 | 9 (56.25) | 7 (43.75) | 0.250 | 10 (62.50) | 6 (37.50) | 0.211 |
| Female | 43 (72.88) | 18 (41.86) | 25 (58.14) | | 21 (48.84) | 22 (51.16) | | 17 (39.53) | 26 (60.47) | | 19 (44.19) | 24 (55.81) | |
| **Age** | | | | | | | | | | | | | |
| Below median (<45 years) | 28 (47.46) | 12 (42.86) | 16 (57.14) | 0.358 | 16 (57.14) | 12 (42.86) | 0.243 | 13 (46.43) | 15 (53.57) | 0.728 | 12 (42.86) | 16 (57.14) | 0.358 |
| Above median | 31 (52.54) | 17 (54.84) | 14 (45.16) | | 13 (41.94) | 18 (58.06) | | 13 (41.94) | 18 (58.06) | | 17 (54.84) | 14 (45.16) | |
| **Marital status** | | | | | | | | | | | | | |
| Married | 44 (74.58) | 19 (43.18) | 25 (56.82) | 0.116 | 19 (43.18) | 25 (56.82) | 0.116 | 18 (40.91) | 26 (59.09) | 0.403 | 20 (45.45) | 24 (54.55) | 0.33 |
| Others [a] | 15 (25.42) | 10 (66.67) | 5 (33.33) | | 10 (66.67) | 5 (33.33) | | 8 (53.33) | 7 (46.67) | | 9 (60.00) | 6 (40.00) | |
| **Education** | | | | | | | | | | | | | |
| Below secondary [b] | 39 (66.10) | 22 (56.41) | 17 (43.59) | 0.119 | 19 (48.72) | 20 (51.28) | 0.926 | 19 (48.72) | 20 (51.28) | 0.315 | 21 (53.85) | 18 (46.15) | 0.314 |
| Secondary and above | 20 (33.90) | 7 (35.00) | 13 (65.00) | | 10 (50.00) | 10 (50.00) | | 7 (35.00) | 13 (65.00) | | 8 (40.00) | 12 (60.00) | |
| **Highest level of household education** | | | | | | | | | | | | | |
| Below secondary [b] | 12 (20.34) | 6 (50.00) | 6 (50.00) | 0.948 | 7 (58.33) | 5 (41.67) | 0.476 | 5 (41.67) | 7 (58.33) | 1.000 [g] | 3 (25.00) | 9 (75.00) | 0.061 |
| Secondary and above | 47 (79.66) | 23 (48.94) | 24 (51.06) | | 22 (46.81) | 25 (53.19) | | 21 (44.68) | 26 (55.32) | | 26 (55.32) | 21 (44.68) | |
| **Religion** | | | | | | | | | | | | | |
| Hindu | 55 (93.22) | 27 (49.09) | 28 (50.91) | 1.000 [g] | 27 (49.09) | 28 (50.91) | 1.000 [g] | 24 (43.64) | 31 (56.36) | 0.805 | 28 (50.91) | 27 (49.06) | 0.612 [g] |
| Others [c] | 4 (6.78) | 2 (50.00) | 2 (50.00) | | 2 (50.00) | 2 (50.00) | | 2 (50.00) | 2 (50.00) | | 1 (25.00) | 3 (75.00) | |
| **Caste** | | | | | | | | | | | | | |
| SC/ST | 34 (57.63) | 13 (38.24) | 21 (61.76) | **0.050** [c] | 15 (44.12) | 19 (55.88) | 0.367 | 12 (35.29) | 22 (64.71) | 0.113 | 13 (38.24) | 21 (61.76) | **0.050** [h] |
| Others (MBC/BC/FC) | 25 (42.37) | 16 (64.00) | 9 (36.00) | | 14 (56.00) | 11 (44.00) | | 14 (56.00) | 11 (44.00) | | 16 (64.00) | 9 (36.00) | |
| **Family type** | | | | | | | | | | | | | |
| Nuclear family | 26 (44.07) | 14 (53.85) | 12 (46.15) | 0.522 | 18 (69.23) | 8 (30.77) | **0.006** [h] | 13 (50.00) | 13 (50.00) | 0.415 | 18 (69.23) | 8 (30.77) | **0.006** [h] |
| Extended family [d] | 33 (55.93) | 15 (45.45) | 18 (54.55) | | 11 (33.33) | 22 (66.67) | | 13 (39.39) | 20 (60.61) | | 11 (33.33) | 22 (66.67) | |
| **Occupation** | | | | | | | | | | | | | |
| Unskilled/Semi-skilled | 37 (67.27) | 21 (51.22) | 20 (48.78) | 0.504 | 20 (48.78) | 21 (51.22) | 0.931 | 22 (53.66) | 19 (46.34) | **0.025** [h] | 19 (46.34) | 22 (53.66) | 0.391 |
| Skilled [e] | 18 (32.73) | 8 (44.44) | 10 (55.56) | | 9 (50.00) | 9 (50.00) | | 4 (22.22) | 14 (77.78) | | 10 (55.56) | 8 (44.44) | |
| **Highest level of household occupation** | | | | | | | | | | | | | |
| Unskilled/Semi-skilled | 48 (87.27) | 23 (44.23) | 29 (55.77) | 0.101 [g] | 25 (48.08) | 27 (51.92) | 0.706 [g] | 24 (46.15) | 28 (53.85) | 0.449 [g] | 25 (48.08) | 27 (51.92) | 0.696 [g] |

*(Continued)*

**Table 1.** (Continued)

| Factors | Total | Attitude | | | Subjective norm | | | Perceived behavioural control | | | Intention to participate | | |
|---|---|---|---|---|---|---|---|---|---|---|---|---|---|
| | n = 59 | Low | High | Chi-Square test | Low | High | Chi-Square test | Low | High | Chi-Square test | Low | High | Chi-Square test |
| | | n = 29 | n = 30 | | n = 29 | n = 30 | | n = 26 | n = 33 | | n = 29 | n = 30 | |
| | n (%) | n (%) | n (%) | p-value | n (%) | n (%) | p-value | n (%) | n (%) | p-value | n (%) | n (%) | p-value |
| Skilled [e] | 7 (12.73) | 6 (85.71) | 1 (14.29) | | 4 (57.14) | 3 (42.86) | | 2 (28.57) | 5 (71.43) | | 4 (57.14) | 3 (42.86) | |
| **Household toilet** | | | | | | | | | | | | | |
| No | 34 (57.63) | 17 (50.00) | 17 (50.00) | 0.879 | 17 (51.52) | 16 (48.48) | 0.683 | 17 (50.00) | 17 (50.00) | 0.284 | 20 (58.82) | 14 (41.18) | 0.083 |
| Yes | 25 (42.37) | 12 (48.00) | 13 (52.00) | | 12 (46.15) | 14 (53.85) | | 9 (36.00) | 16 (64.00) | | 9 (36.00) | 16 (64.00) | |
| **Member of a social group [f]** | | | | | | | | | | | | | |
| No | 33 (55.93) | 16 (48.48) | 17 (51.52) | 0.908 | 17 (51.52) | 16 (48.48) | 0.318 | 13 (39.39) | 20 (60.61) | 0.415 | 18 (54.55) | 15 (45.45) | 0.351 |
| Yes | 26 (44.07) | 13 (50.00) | 13 (50.00) | | 10 (38.46) | 16 (61.54) | | 13 (50.00) | 13 (50.00) | | 11 (42.31) | 15 (57.69) | |
| **Participated in DeWorm3 community sensitisation activities** | | | | | | | | | | | | | |
| No | 30 (50.85) | 17 (56.67) | 13 (43.33) | 0.240 | 19 (63.33) | 11 (36.67) | 0.027 [h] | 15 (50.00) | 15 (50.00) | 0.351 | 16 (53.33) | 14 (46.67) | 0.514 |
| Yes | 29 (49.15) | 12 (41.38) | 17 (58.62) | | 10 (34.48) | 19 (65.52) | | 11 (37.93) | 18 (62.07) | | 13 (44.83) | 16 (55.17) | |
| **Familiar with community drug distributor** | | | | | | | | | | | | | |
| No | 7 (11.86) | 4 (57.14) | 3 (42.86) | 0.706 [g] | 4 (57.14) | 3 (42.86) | 0.652 | 3 (42.86) | 4 (57.14) | 1.000 [g] | 3 (42.86) | 4 (57.14) | 1.000 [g] |
| Yes | 52 (88.14) | 25 (48.08) | 27 (51.92) | | 25 (48.08) | 27 (51.92) | | 23 (44.23) | 29 (55.77) | | 26 (50.00) | 26 (50.00) | |
| **Perceived severity of the adverse event** | | | | | | | | | | | | | |
| Severe | 34 (57.63) | 16 (47.06) | 18 (52.94) | 0.708 | 19 (55.88) | 15 (44.12) | 0.228 | 17 (50.00) | 17 (50.00) | 0.284 | 16 (47.06) | 18 (52.94) | 0.708 |
| Moderate/mild | 25 (42.37) | 13 (52.00) | 12 (48.00) | | 10 (40.00) | 15 (60.00) | | 9 (36.00) | 16 (64.00) | | 13 (52.00) | 12 (48.00) | |

[a] Others–unmarried/widowed/separated

[b] Below grade 10

[c] Others–Christians/Muslims

[d] three or more generations living together or living with relatives

[e] Skilled–tailoring/carpentry/welding/mechanics

[f] Social group (community-based groups)–women's self-help groups/farmers' club/cooperative societies

[g] Fisher's exact test for factors with less than 5 expected cell counts

[h] Significant at p-value ≤0.05

Acronyms: SC/ST-Scheduled Caste/Scheduled Tribe, MBC/BC/FC–Most Backward Class/Backward Class/Forward Class

### Individuals engaged in skilled occupations had higher perceived behavioural control in participating in cMDAs

Behavioural control reflects individuals' self-efficacy to engage in cMDA. Most participants were confident that they would be able to access deworming tablets distributed during cMDA campaigns if they wanted to (90%), and they would participate in future STH cMDAs even if their neighbours and friends tell them not to participate (78%). Many of them (76%) reported that buying deworming tablets from private medical stores for their family members was prohibitively expensive. Most participants reported that they would participate in future STH cMDAs (85%) and would not wait for approval from any family member (61%). In-depth

interviews confirmed that most individuals feel free to independently make decisions about their own treatment, while a smaller minority are influenced by family members,

> "*First time when they gave the tablet got diarrhoea. . . next time my family also said not to eat. . . I ate a tablet the second time. . . It is not that I will eat only when my family members tell me to eat. It is my choice.*" (Woman #1- perceived moderately severe AE, Cluster 38)

> "*I will eat. My son said, 'ma, you eat the tablet. On that day, something unfortunate happened.' Then I must listen to my eldest son. When he tells, I will think about what my son had said.*" (Woman #1- perceived moderately severe AE, Cluster 24)

Overall, 56% of the participants had a higher perceived behavioural control (≥median score of 25; maximum score–30) to participate in cMDA. Perceived behavioural control was significantly associated with participants' occupation ($\chi^2$ = 5.02, $P<0.05$) (Table 1). Those engaged in skilled occupations (78%) such as tailoring, carpentry, welding, mechanics etc. had higher perceived behavioural control than those engaged in unskilled/semi-skilled occupations such as daily labourers, peons, and security guards (46%).

### Intention to participate in future cMDAs was high among the disadvantaged socio-economic groups (scheduled caste/scheduled tribe) and extended families

A majority of participants (85%) said they would consume or let their child consume the tablet in the future, although one-third (32%) believed that the tablet could cause an AE and were concerned about it for themselves and their family members. The in-depth interviews showed that the individuals' decision to participate was based on the perceived health benefit despite experiencing an AE because deworming relieved them of stomach pain, increased their appetite, and they felt more energetic to perform their work. They said,

> "*After eating the tablet I was able to move quickly from place to place, my body was active. . .. I do not have intestinal worms so I think I am healthy. Once in six months I eat tablet. My menstrual cycle is regular when I eat the tablet.*" (Woman #1- perceived moderately severe AE, Cluster 23)

> "*It is good to eat the tablet, Sir. We feel well after eating the tablet, we do not have intestinal worms. Where I work, many ladies from this village and the neighbouring village come, they also say that their body feels well after eating the tablet.*" (Woman #1- perceived moderately severe AE, Cluster 12)

Overall, 51% of the participants had a high intention to participate in future cMDAs (≥median score of 18; maximum score–20). A high level of intention to participate was significantly associated with caste and family type ($\chi^2$ = 3.83, $P$ = 0.050 and $\chi^2$ = 7.50, $P<0.05$ respectively) (Table 1). Intention to participate was higher among those who belonged to SC/ST (62%) than those who belonged to other castes (36%) and among those belonging to extended families (67%) compared to nuclear families (31%).

### Attitude and perceived behavioural control were significant predictors of intention to participate

The framework of TPB wherein attitude, subjective norm, and perceived behavioural control are hypothesised to have a relationship with intention to participate in cMDA was used to

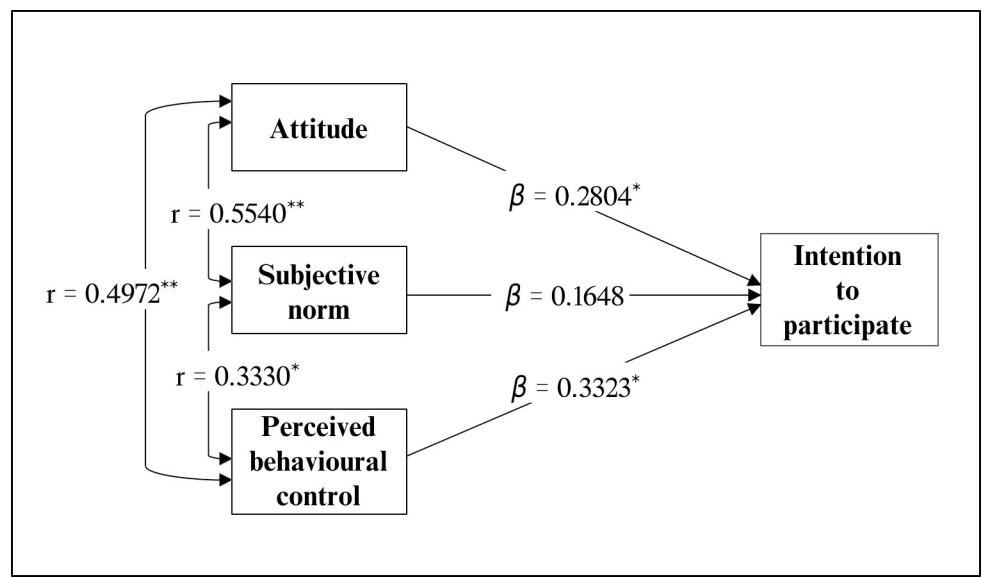

**Fig 3. Path model–the relationship between attitude, subjective norm, perceived behavioural control with intention to participate.** Note: All parameter values are standardised; $^*p < .05$, $^{**}p < .001$; r–correlation coefficients; β–regression coefficients, the outcome variable increases β times for every unit increase in a predictor variable.

inform a path analysis (Fig 3). The path model showed a good fit for the data ($\chi2 = 29.80$, $P< 0.001$), accounting for 39.7% variance in intention to participate. Attitude and perceived behavioural control (composite scores) were significant predictors of future intention to participate in cMDA. The mean value of intention to participate increased by 0.28 for every unit change in attitude and by 0.33 for every unit change in perceived behavioural control. Subjective norms did not emerge as a significant predictor of intention to participate. The analysis also showed a significantly strong relationship between the independent variables of attitude, subjective norm and perceived behavioural control.

## Discussion

This paper examines the perspectives of those who reported an AE after taking albendazole during cMDA for STH and investigates the behavioural factors that influence their intention to participate in future rounds of deworming. In this study, more women than men and children reported experiencing AEs during MDA as also been observed in other countries as well as India [34]. Participation in cMDA after an AE in this study was high, with only 12% refusing to participate in the next cMDA and only 4% refusing to participate in all subsequent cMDA rounds. These rates are similar to other studies that have evaluated the impact of AEs on participation in MDA programs; LF study in Ghana, trachoma study in Ethiopia and malaria study in Gambia have reported refusal to participate in MDA ranging between 7–33% as a result of individuals having experienced an AE [35–37]. Intention to participate in future cMDA was also significantly higher among those who belonged to SC/ST (a caste group officially designated the most disadvantaged socio-economic groups in India), and extended families, possibly because of the high perceived risk of STH infection and family support to participate in cMDAs, as shown in the in-depth interviews.

Attitude and perceived behavioural control were significant predictors of intention to participate in future STH cMDA. Despite experiencing an AE, most participants believed that the tablet was safe and effective probably because they considered deworming to be good for their

health [38]. A positive attitude ($\geq$ median composite score) towards cMDA was significantly higher among the most disadvantaged households who belonged to SC/ST. Another earlier study by our team demonstrated that a higher proportion of CDDs who belong to SC/ST households have a positive attitude towards cMDA [39]. The likelihood of perceived risk to STH infections may be higher among SC/ST households as they have poor water and sanitation facilities and less access to healthcare [40]. Overall, half of the participants believed that STH is a minor infection (49%), and deworming was not necessary for those who use toilets (53%). Therefore, messages on STH-related morbidities and the importance of high treatment coverage for STH elimination need to be incorporated into community sensitisation activities. Individuals engaged in skilled occupations were found to have higher perceived behavioural control in participating in cMDA, possibly due to increased agency as a result of higher levels of freedom of movement, decision-making capacity, and income. A Sri Lankan study on LF MDA program also showed that participation was higher among skilled workers in the middle-income group as compared to those in lower- and upper-income groups [41].

Subjective norm values, or the individual's perceived social expectations, were significantly higher among individuals with strong ties to family and community. These individuals are more likely to believe that those around them are participating in cMDA. This may have been shaped by observing community members' responses and expectations during community sensitisation activities and the intrahousehold discussion about community norms concerning cMDAs. However, path analysis showed no significant relationship between subjective norm and intention to participate. Analysis of two studies on voting and energy consumption that examined the interactions between attitude, subjective norm, and perceived behavioural control on the prediction of intention showed that higher perceived behavioural control tends to strengthen the attitude and weakens subjective norm in the prediction of intention [42]. This study also showed a significantly higher correlation of perceived behavioural control with attitude than with subjective norms.

Participation in subsequent cMDAs after an AE was high in our study. However, about two-thirds of the participants in our study were concerned about the occurrence of an AE in the future. The in-depth interviews also revealed that when the AE is perceived as being severe, it could even lead to household-level refusal or refusal to treat their children. Such non-compliers may be a source of STH infection, as observed in an LF-MDA study in Haiti [43]. Community members in an LF-MDA study in India suggested that CDDs should be given training and drugs to manage AEs as concerns about AEs persisted even though AEs reduced over the course of repeated MDAs [15]. Our qualitative research also demonstrates that CDDs were often the first point of contact for community members when they experienced the AE, therefore, MDA programs will benefit from CDDs providing counselling to community members to reduce fear, particularly among those experiencing AEs. Adopting a door-step treatment delivery strategy provides an opportunity to counsel those who fear an AE. Messages on the management of common side-effects may be helpful as about 10% of the non-compliers in an LF-MDA study from Odisha in India reported that they would have complied if they knew how to manage the side-effects [44]. Incorporating messages in community sensitization activities on AEs, that the drug is safe, and the side-effects are short-term as well as about access to care in case of an AE, and management of AE by qualified health personnel in the community may be critical to optimize successful implementation of MDA programs with high coverage [45,46].

## Strengths and limitations

This study had several notable strengths. The surveys and interviews were conducted within the context of a highly rigorous randomized trial in which detailed information allowed the

identification of all individuals with reported AEs. Participation rates were high and the repeated cMDA rounds within the parent trial allowed for an assessment of behaviour over time. However, this analysis also had limitations. AE surveillance was conducted passively, so it is possible that some AEs of mild or even moderate severity may not have been reported. However, the longitudinal data collected at the individual level provided an opportunity to observe how a reported AE affects participation in subsequent cMDAs. Measuring only the intention to participate in future cMDAs instead of actual behaviour was not possible in this study because the trial included only six rounds of cMDA. Additionally, there could be a recall bias as we carried out this research six months after completing all six rounds of cMDA due to delays driven by the COVID-19 pandemic. The findings of this study may not be generalisable to other geographies and other NTDs where similar studies would be required.

## Conclusion

This study demonstrates that positive attitude and perceived behavioural control are significantly associated with the intention to participate in a future round of cMDA, even after experiencing an AE. Caste (SC/ST households), family type (extended families), occupation (engaged in skilled occupation), and participation in community sensitisation activities were factors significantly associated with various TPB constructs. Messages on AE and its management need to be incorporated in community mobilization activities and specifically aimed at castes other than SC/ST and nuclear families in MDA programs. CDDs involved in MDA programs need to be oriented about AEs during cMDAs as they are important stakeholders in mitigating the fear of AEs. In addition, research using the TPB framework with larger sample sizes would better inform about behavioural factors influencing intention to participate after an AE in STH MDA and other NTD programs, particularly those transitioning from control to elimination programs.

## Supporting information

**S1 File. Strengthening the Reporting of Observational Studies in Epidemiology (STROBE) checklist.**
(DOCX)

**S2 File. Survey questionnaire and qualitative question guide, AE study.**
(DOCX)

## Acknowledgments

The authors wish to thank all of the study participants, communities and community leaders, national NTD program staff and other local, regional and national partners who have participated or supported the DeWorm3 study. We also would like to acknowledge the work of all members of the DeWorm3 study teams and affiliated institutions. We give special thanks to Isaac Arputharangam, and Prasanna Yesuratinam for their support in data collection, and the field managers, Rajeshkumar Rajendiran and Chinnaduraipandi Paulsamy, for providing logistical support in the field. We thank Naveenkumar Sekar for creating the data entry form of the questionnaire and Uma Maheshwari for her support in transcribing qualitative audio files.

## Author Contributions

**Conceptualization:** Kumudha Aruldas, Arianna Rubin Means, Sitara S. R. Ajjampur.

**Data curation:** Gideon John Israel, Jabaselvi Johnson, Angelin Titus, Malvika Saxena.

**Formal analysis:** Kumudha Aruldas, Gideon John Israel, Saravanakumar Puthupalayam Kaliappan, Rohan Michael Ramesh.

**Funding acquisition:** Judd L. Walson.

**Investigation:** Kumudha Aruldas, Jabaselvi Johnson, Angelin Titus.

**Methodology:** Kumudha Aruldas, Arianna Rubin Means.

**Project administration:** Sitara S. R. Ajjampur.

**Resources:** Sitara S. R. Ajjampur.

**Supervision:** Kumudha Aruldas.

**Visualization:** Kumudha Aruldas, Gideon John Israel.

**Writing – original draft:** Kumudha Aruldas, Gideon John Israel.

**Writing – review & editing:** Kumudha Aruldas, Gideon John Israel, Jabaselvi Johnson, Angelin Titus, Malvika Saxena, Saravanakumar Puthupalayam Kaliappan, Rohan Michael Ramesh, Judd L. Walson, Arianna Rubin Means, Sitara S. R. Ajjampur.

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
