## [Decision Letter · Decision Letter 0]

12 Dec 2022

Dear Prof. Ajjampur,

Thank you very much for submitting your manuscript "Impact of adverse events during community-wide mass drug administration for soil-transmitted helminths on subsequent participation – a Theory of Planned Behaviour analysis" for consideration at PLOS Neglected Tropical Diseases. As with all papers reviewed by the journal, your manuscript was reviewed by members of the editorial board and by several independent reviewers. In light of the reviews (below this email), we would like to invite the resubmission of a significantly-revised version that takes into account the reviewers' comments. 

We cannot make any decision about publication until we have seen the revised manuscript and your response to the reviewers' comments. Your revised manuscript is also likely to be sent to reviewers for further evaluation.

Sincerely,

Georgios Pappas

Academic Editor

Eva Clark

Section Editor

Please respond to the reviewer comments

Reviewer's Responses to Questions

**Key Review Criteria Required for Acceptance?**

**Methods**

-Are the objectives of the study clearly articulated with a clear testable hypothesis stated?

-Is the study design appropriate to address the stated objectives?

-Is the population clearly described and appropriate for the hypothesis being tested?

-Is the sample size sufficient to ensure adequate power to address the hypothesis being tested?

-Were correct statistical analysis used to support conclusions?

-Are there concerns about ethical or regulatory requirements being met?

Reviewer #1: The objectives of the study are clearly articulated in the paper with a testable hypothesis stated. The study design is appropriate and addressed the stated objectives as well. The study population is well-described and appropriate for the hypothesis being tested. As stated, though the statistical analysis was difficult to understand in certain portions the due to obvious limitations of one’s knowledge and understanding of statistics, the conclusions and recommendations are well aligned with the study results and therefore outcomes. I do not have any ethical or regulatory concerns regarding this study.

Reviewer #2: The required corrections are written in detail in the article file.

The purpose of the study, statistical population and data sampling and analysis are written in the required corrections article

Reviewer #3: The article looks at Impact of adverse events during community-wide mass drug administration for STH , under the DeWorm3 trial site in southern India using the Theory of Planned Behaviour (TPB), to understand perceptions about participating in cMDA for STH and drivers of participation in subsequent STH cMDA programs after experiencing an adverse effect. The objectives are clearly outlined.They have used a multi-method approach was used, including quantitative and qualitative data collection, which is appropriate for this study. The study population is clearly defined. Sample size involved all with adverse effects .Analysis was appropriately done.

No issues on methods.

Reviewer #4: (No Response)

**Results**

-Does the analysis presented match the analysis plan?

-Are the results clearly and completely presented?

-Are the figures (Tables, Images) of sufficient quality for clarity?

Reviewer #1: The analysis presented clearly matches the analysis plan and the results are clearly presented. The figures are of sufficient quality and clarify though the repetition of certain charts is unclear to me. 

The findings of this study though not entirely novel are deeply insightful. Of significance is the influence of education, social status, and occupation on the intention to participate in future cMDA after experiencing AEs. It is noteworthy that in other MDA surveys, compliance among the educated tends to be lower than among the uneducated as demonstrated by this study. The sub-topics in the results section, make the concept, hypothesis and results easy to understand and follow. 

The magnitude of the challenge of AEs during MDAs and particularly in albendazole treatment requires more information in the paper. Another phenomenon experienced during MDAs is the issue of side events that cannot be attributed to MDA but rather a co-incidental event often detected upon investigations. So the question of efforts made to isolate actual events of AEs from coincidental events needs to be clarified including the understanding the timing of the AEs post-treatment.

Reviewer #2: Corrections related to the results section are written in the required corrections article

Reviewer #3: results are clearly presented but 

Line 238 Fig 1 . the figure is not clear presented; participated and refused are not clear shown ; this can be improved . There are also some graphs and comments in the table (I think it was not finalized) 

Line 323: what about positive attitude association with family type (nuclear or extended)?.

Reviewer #4: (No Response)

**Conclusions**

-Are the conclusions supported by the data presented?

-Are the limitations of analysis clearly described?

-Do the authors discuss how these data can be helpful to advance our understanding of the topic under study?

-Is public health relevance addressed?

Reviewer #1: The conclusions are supported by the data presented with clearly stated limitations of the study. Discussions on how these data can be helpful to advance our understanding of the topic under study can however be made stronger with further justification on its utility to the global program. The study is definitely of significant public health relevance in informing the trajectory of the transition from control to elimination of soil transmitted helminthes.

Reviewer #2: Corrections related to the conclusion section are written in the required corrections article

Reviewer #3: This is clearly supported by the data presented. However, recall bias may have contributed to a lot of changes in these findings as 6 months after MDA is a long time to remember. Is it possible that there were no gender differences in attitudes due to recall bias.

Reviewer #4: (No Response)

**Editorial and Data Presentation Modifications?**

Reviewer #1: Very useful conclusions and recommendations are made in the paper. Though these conclusions and recommendations are not entirely new they emphasize the utility of TPH in studying the three aspects of participants’ attitudes, subjective norms, and perceived behaviour control and an individual’s intention to participate in cMDA. This a Remarkable paper and should be published with or without the suggested modifications.

Reviewer #2: Corrections are written in the required corrections article

Reviewer #3: Figure 1 need to be reworked

Reviewer #4: (No Response)

**Summary and General Comments**

Reviewer #1: This manuscript should be of great interest to the global NTD community considering the inflection point of the global program with the current discussion and anticipated change from control to elimination. The impact of adverse events on control/elimination activities including mass drug administration is particularly critical. Furthermore, community-wide MDA as against school-based MDA has been recommended as the path forward towards the elimination of soil-transmitted helminthiasis. Very useful conclusions and recommendations are made in the paper. Though these conclusions and recommendations are not entirely new they emphasize the utility of TPH in studying the three aspects of participants’ attitudes, subjective norms, and perceived behaviour control and an individual’s intention to participate in cMDA. This a Remarkable paper and should be published with or without the suggested modifications.

Reviewer #2: Corrections are written in the required corrections article

Reviewer #3: This was a good study , but it was mainly limited by the long period between the last MDA and the actual research . Some of the findings are likely to be affected by recall bias. Nether the less there is still some good information obtained from this study

Reviewer #4: Dear Authors

Here my comments:

Line 188-189: Please indicate here when Chi-square test or Fischer exact test were used or as footnote in the result table. 

Line 190: Why call the four the TPB variables outcome variable here? Given the descriptive nature of chi-Square test, it is preferable to call them the dependent variables. As mentioned in Line 196-197 the intention variable is the outcome variable. 

Line 190: Not necessary to quality path analysis as “multiple regression-based”…… To simply say path analysis with what it does is sufficient.

Line 192: “This determines…. What is “This” ? Please rephrase the sentence to make clearer. 

Line 199: How as G-power used to calculate the power of a statistical test? What is that “a” statistical test you used it for in your study? There is a need to have clear procedure here. What was used? Why and how?

I suggest you give a clear information here under data analysis section, how the minimum sample size required for the path analysis was estimated with references, rather giving a post-hoc power analysis as shown in your result section. This is important given the small sample size (59) used in your path analysis. 

Line 202: ,”final codes arrived with consensus”,… Please check grammar.

Line 420-423. Please refer to my earlier comments on sample size. This information should be in data analysis section Please state the names of your predictors. And the reference/justification for the choice of your effect size of 0.35. 

Strengths and limitations

There is a need to mention limitation regarding lack of generalizability of the study findings given the sample size and been part of a randomized trial with usually have a set of criteria that limits the inclusion of a section of the wider population.
---

## [Decision Letter · Decision Letter 1]

6 Feb 2023

Dear Prof. Ajjampur,

Thank you very much for submitting your manuscript "Impact of adverse events during community-wide mass drug administration for soil-transmitted helminths on subsequent participation – a Theory of Planned Behaviour analysis" for consideration at PLOS Neglected Tropical Diseases. As with all papers reviewed by the journal, your manuscript was reviewed by members of the editorial board and by several independent reviewers. The reviewers appreciated the attention to an important topic. Based on the reviews, we are likely to accept this manuscript for publication, providing that you modify the manuscript according to the review recommendations - some minor improvements are still needed.

Sincerely,

Georgios Pappas

Academic Editor

Eva Clark

Section Editor

some minor improvements needed

Reviewer's Responses to Questions

**Key Review Criteria Required for Acceptance?**

**Methods**

-Are the objectives of the study clearly articulated with a clear testable hypothesis stated?

-Is the study design appropriate to address the stated objectives?

-Is the population clearly described and appropriate for the hypothesis being tested?

-Is the sample size sufficient to ensure adequate power to address the hypothesis being tested?

-Were correct statistical analysis used to support conclusions?

-Are there concerns about ethical or regulatory requirements being met?

Reviewer #2: In the introduction section, the abbreviation can be written as a subtitle, but it must be according to the guidelines of the journal.

In the introduction section, the reason for using this theory should be explained more clearly in the introduction section of the study.

The code of study ethics should be written

Mention how to comply with the ethical standards and the code of ethics in the work method

**Results**

-Does the analysis presented match the analysis plan?

-Are the results clearly and completely presented?

-Are the figures (Tables, Images) of sufficient quality for clarity?

Reviewer #2: (No Response)

**Conclusions**

-Are the conclusions supported by the data presented?

-Are the limitations of analysis clearly described?

-Do the authors discuss how these data can be helpful to advance our understanding of the topic under study?

-Is public health relevance addressed?

Reviewer #2: Limitations of the study and strengths and weaknesses should also be corrected.

Many corrections requested by the referees have been modified by the authors

**Editorial and Data Presentation Modifications?**

Reviewer #2: (No Response)

**Summary and General Comments**

Reviewer #2: (No Response)

PLOS authors have the option to publish the peer review history of their article (what does this mean?). If published, this will include your full peer review and any attached files.

Reviewer #2: No

Figure Files:

Data Requirements:

Reproducibility:

References

---

## [Editor Report · Decision Letter 2]

7 Feb 2023

Dear Prof. Ajjampur,

We are pleased to inform you that your manuscript 'Impact of adverse events during community-wide mass drug administration for soil-transmitted helminths on subsequent participation – a Theory of Planned Behaviour analysis' has been provisionally accepted for publication in PLOS Neglected Tropical Diseases.

Best regards,

Georgios Pappas

Academic Editor

Eva Clark

Section Editor

---

## [Editor Report · Acceptance letter]

21 Feb 2023

Dear Prof. Ajjampur,

We are delighted to inform you that your manuscript, "Impact of adverse events during community-wide mass drug administration for soil-transmitted helminths on subsequent participation – a Theory of Planned Behaviour analysis," has been formally accepted for publication in PLOS Neglected Tropical Diseases.

Best regards,

Shaden Kamhawi

co-Editor-in-Chief

Paul Brindley

co-Editor-in-Chief
